# Deep Neural Network for 3D Shape Classification Based on Mesh Feature

**DOI:** 10.3390/s22187040

**Published:** 2022-09-17

**Authors:** Mengran Gao, Ningjun Ruan, Junpeng Shi, Wanli Zhou

**Affiliations:** 1School of Electronic and Information, Yangtze University, Jingzhou 434023, China; 2College of Electronic Countermeasure, National University of Defense Technology, Hefei 230037, China

**Keywords:** triangular mesh, graph convolutional neural networks, 3D shape classification

## Abstract

Virtual reality, driverless cars, and robotics all make extensive use of 3D shape classification. One of the most popular ways to represent 3D data is with polygonal meshes. In particular, triangular mesh is frequently employed. A triangular mesh has more features than 3D data formats such as voxels, multi-views, and point clouds. The current challenge is to fully utilize and extract useful information from mesh data. In this paper, a 3D shape classification network based on triangular mesh and graph convolutional neural networks was suggested. The triangular face of this model was viewed as a unit. By obtaining an adjacency matrix from mesh data, graph convolutional neural networks can be utilized to process mesh data. The studies were performed on the ModelNet40 dataset with an accuracy of 91.0%, demonstrating that the classification network in this research may produce effective results.

## 1. Introduction

With the rapid development of 3D data-capturing devices, 3D data collection has become more convenient and faster. Three-dimensional data are foundational to computer graphics and computer vision, and it contains a wealth of geometric, shape, and scale information. The use of 3D models is increasing in daily life, such as in autonomous driving [1,2], virtual reality, and remote sensing mapping [3], all of which require advanced processing [4] and analysis of the collected 3D data. The study of how to effectively classify, identify, and segment 3D models is a hot topic at present.

As computer vision and deep learning have developed rapidly, the study of 3D shapes has shifted from handcraft features [5,6] to deep learning methods. Three-dimensional shapes are available in various representations such as voxels, multi-view, point cloud and polygon meshes, as shown in Figure 1. Early convolutional neural networks have made great strides in the classification of 2D images, but they cannot be applied to unstructured data. MVCNN [7] used the deep learning algorithm on 2D images under a multi-view of the 3D model and obtained the classification results of 3D shapes. However, the multi-view lacked depth information and could not describe the 3D spatial characteristics of objects well. Recently, graph convolutional neural networks have achieved good results on point cloud classification tasks. DGCNN [8] uses the edge convolution module to learn topological features of point clouds by graph convolutional networks, which can better capture local geometric information and achieve better results, but with too many training parameters.

Voxel, multi-view, and point cloud are currently used more frequently than mesh data in most 3D model classification techniques. The point cloud lacks surface information on the objects and is disorganized, having few connections between its points. Information will unavoidably be lost during the voxelization process of 3D models, and large resolutions will necessitate large amounts of memory. Issues with feature learning based on multi-view and voxel data include significant 3D feature loss and numerous processing stages. Mesh data include information about features including vertices, edges, faces, corners, and normals. Mesh data are more descriptive of objects than other types of data, and it is now being investigated how to effectively extract information from mesh data and to fully utilize it. Based on the advantage of graph convolutional neural networks [9] excelling at processing non-Euclidean structure data, this paper proposes a 3D model classification network based on graph convolutional neural networks combined with triangular mesh, which can learn the features of 3D models directly and achieve the classification of 3D models. The ModelNet40 [10] dataset is used to test the model, and the results for 3D model classification are promising.

## 2. Related Works

Several methods have been proposed for obtaining features in mesh data. Jiao et al. [11] used several metrics such as dihedral angles, edge angles, and eigenvectors of the vertex normal tensor matrix to identify feature point, feature edge, and feature edge directions. Kim et al. [12,13] calculated the maximum point set to represent the feature points of the 3D mesh by using the average geodesic distance function. Hu et al. [14] proposed a global rarity to describe global saliency: the rarer the part on the 3D mesh, the better the characteristics of the entire mesh.

The following is an introduction to related work on 3D model classification using deep learning. 

Multi-view: Initially, scholars represented the projections of the model from different perspectives as 2D images and then used convolutional neural networks to process the projected images. The MVCNN [7], GVCNN [15], and VMVCNN [16] methods transform the 3D model into 2D images obtained by shooting in different view cases and extracting image features to perform the classification. Zhang et al. [17] proposed an effective recognition model based on multi-view convolutional neural networks. Qi et al. [16] added azimuth and elevation angle change features to the training set to improve the performance of the classification model based on the MVCC. Gao et al. [18] systematically evaluated the performance of deep learning features in view-based 3D model retrieval. Projection from multiple angles leads to a lack of global information perception and is difficult to apply in scene segmentation and object detection tasks.

Volumetric: A volumetric occupancy grid is often used in order to represent the environment state as a 3D mesh. Zhang et al. [19] proposed to convert point clouds into regular voxels and to then input them into convolutional neural networks for feature extraction. Riegler et al. [20] transformed the point cloud into an octree format to extract features. Maturana et al. [21] proposed VoxNet, which incorporates voxels and multi-views. Wu et al. [10] developed a 3D-ShapeNets model that handles voxelization directly. Kd-Net [22] constructed the point cloud into a kd-tree and then classified the points. The spherical CNN [23] projects 3D meshes onto closed spheres, confirming spherical convolution’s effectiveness in point cloud classification. In comparison with the original model, the voxelized model has a low resolution, which leads to a significant loss of information.

Point clouds: PointNet [24] and PointNet++ [25] network models directly process point cloud data. PointNet can learn from disordered point clouds but ignores the extraction of local features. PointNet++ considers local features of point clouds but ignores the connection between points, and the operation of aggregating point set features is time-consuming. Li et al. [26] proposed PointCNN so that the input order of the point cloud does not affect the convolution operation.

Graph: According to the definition [9,27] of convolution in the graph, this calculation method [28,29] has been employed on point clouds. Simonovsky et al. [30] proposed a network edge ECC applied to any graph structure using maximum sampling to aggregate vertex information. However, there is a problem of high computational cost. KCNet [31] orders local point clouds using graphs and aggregates local features through graph convolution. DGCNN [8] provides a dynamic graph edge convolution module that improves the network’s ability to obtain local features, ignoring the vector direction between points, resulting in some information loss. Point GCN [32] recycled edges of the graph within each layer, avoiding unnecessary point cloud grouping and sampling. Grid-GCN [33] proposed a module to reduce theoretical time complexity and to enhance space coverage.

Based on the excellent performance of graph convolutional neural networks in processing unstructured data, in this paper, we propose a 3D shape classification network that combines the features of mesh data. The 3D model is represented as a graph in the model and leverages graph convolutional networks to enhance local feature extraction. The next section describes in detail the model proposed in this paper.

## 3. Methods

### 3.1. Overview

We propose a 3D shape classification network based on triangular mesh features and graph convolutional neural networks. Figure 2 shows the flow chart of the algorithm proposed in this paper. The flow chart consists of two parts, one is the simplification and feature extraction of the mesh data, and the other is the 3D shape classification based on graph convolutional neural networks.

Triangle meshes are a common way to display 3D models, which consist of three parts: vertices, edges, and faces. A face refers to a triangular face formed by interconnecting three adjacent vertices in the mesh data. The triangular mesh data can be defined as M=(V,F), where $V=\left\{Vi|Vi\in R^{3}\right\}$ denotes the set of points and $F=\left\{f_{ijk}=\left(v_{i},v_{j},v_{k}\right)\in V,i\neq j,j\neq k\right\}$ denotes the set of faces. Triangular mesh data are better equipped to describe 3D models than other data types such as voxels, multi-views, and point clouds. Furthermore, the explicit connection feature of the mesh makes it easier to extract the adjacency matrix of the mesh.

We should simplify the original input 3D model mesh data to obtain a model with no more than 1024 faces. In order to combine the data from nearby vertices, the face is supplied into the model. Multi-scale local feature splicing and pooling then produces the classification results. The model for mesh processing and classification is described in full below.

### 3.2. Processing Mesh

The triangular mesh contains a rich set of features that can effectively represent the geometric environment. We will extract three initial features from the mesh as the input to the classification model. The three initial features are shown in Figure 3.

We presumed that the meshes representing the 3D models are manifolds [34], that each edge in the mesh is only connected to one or two faces, and that each triangular face is connected to no more than three other triangular faces. We transform the mesh data into a list of faces, treating the face as the sole unit. The list can be used to find each face’s adjacency index. A face is filled with its own index if it connects with less than three other faces.

Because the coordinates of the three vertices of the face are known to be A, B, and C in the original data, we can obtain the coordinates of the center point of the face and the corner by using Equations (1) and (2).
(1)$$O=\{\begin{array}{c} \begin{array}{c} x_{O}=(x_{A}+x_{B}+x_{C})\times \frac{1}{3}\\ y_{O}=(y_{A}+y_{B}+y_{C})\times \frac{1}{3} \end{array}\\ z_{O}=(z_{A}+z_{B}+z_{C})\times \frac{1}{3} \end{array}$$
(2)$$e_{OA}=\{\begin{array}{c} \begin{array}{c} x_{e}=(x_{A}-x_{O})\\ y_{e}=(y_{A}-y_{O}) \end{array}\\ z_{e}=(z_{A}-z_{O}) \end{array}$$

### 3.3. Model Design

The graph convolutional neural networks are capable of processing unstructured data. The graph features will be learned using the spectral domain graph convolutional neural network proposed by Kipf [9] in this paper. Supported by spectral graph theory, the kernel of the neural network is defined by a filter for graph signal processing, giving it a better filtering capability. The input of the graph convolutional network consists of two parts: First, a feature description $x_{i}$ for each node i, which denotes the N×M feature matrix (N denotes the number of nodes, and M denotes the number of features of the input). The second is the adjacency matrix of the graph. (Construct the adjacency matrix using the adjacency relationship between faces.) The value of each element in the adjacency matrix can be calculated by Equation (3).
(3)$$A[i,j]=\left\{ \begin{array}{l} \begin{array}{cc} 1, & \left(v_{i},v_{j})\in E(G\right) \end{array}\\ \begin{array}{cc} 0, & \left(v_{i},v_{j})\notin E(G\right) \end{array} \end{array}\right.$$

Suppose that there are L layers of graph convolution and that l denotes the current number of layers. $H^{\left(l\right)}$ denotes the output of layer
l. A. denotes the adjacency matrix of the graph of N nodes ($A\notin R^{N\times N}$), so each graph convolutional layer neural network can be represented by the nonlinear function Equation (4).
(4)$$H^{\left(l+1\right)}=f(H^{l},A)$$

For a graph convolutional neural network to retain information about the nodes, each node needs to be connected to itself. Next, normalize A by $D^{-1/2}AD^{-1/2}$, where D is the degree diagonal matrix of the nodes and $A=A+I_{N}$ ($I_{N}$ is the unit matrix). The propagation equation of the graph convolutional neural network can be expressed as
(5)$$H^{\left(l+1\right)}=\sigma \left(D^{-\frac{1}{2}}AD^{-\frac{1}{2}}H^{\left(l\right)}W^{\left(l\right)}\right)$$

Equation (5) shows that constructing the graph’s adjacency matrix is the key to using graph convolution. In this paper, the adjacency matrix and the degree matrix can be built by the connection relationship with the face. The graph convolution can be applied to the mesh data.

Before inputting the face features into the graph convolutional network for aggregation, the center point features and the corner vector features need to be processed separately. Referring to the method in MeshNet [35], rotational convolution is used to process the corner vectors of the triangular face, which only works on two corner vectors at a time. Figure 4 shows the diagram of rotational convolution. Suppose that a, b, c are the vectors of a face from the center point to the three angles and define its convolution output as Equation (6). Finally, the output is passed through the fully connected layer to obtain a feature with a length of 64.
(6)a⊗b+b⊗c+c⊗a
where ⊗ means a convolution operation and $\left(a,b,c\right)\in R^{3}$.

The $O\in R^{N\times 3}$ denotes the center point feature of the face. Increasing the center point feature to 64 dimensions through a fully connected layer results in obtaining the center point feature dimension as $O\in R^{N\times 64}$. We connect the convolutional output of the corner vectors with the features of the center points to obtain the high-dimensional features and to input them into the graph convolutional neural network for feature aggregation.

In this paper, a two-layer graph convolutional neural network is used. We input the high-dimensional features into the first layer of graph convolution to obtain the information on aggregated first-degree neighboring vertices. Then, after the activation function, the output features go through the second layer of graph convolution to obtain the information of the aggregated second-degree neighboring vertices. The information of the aggregated first-degree neighboring vertices and the aggregated second-degree neighboring vertices are stitched together to obtain multi-resolution features, as shown in Figure 5. Degree = 1 denotes the aggregation of first-degree neighboring vertices, and degree = 2 denotes the aggregation of second-degree neighboring vertices.

The multi-resolution features are passed through a fully connected layer (1024) to obtain higher-dimensional features, and the information redundancy in the high-dimensional space facilitates the subsequent global pooling operation. In order to obtain the global features, global fusion of the extracted features is required. Max pooling is a nonlinear feature fusion function that is insensitive to the order of elements, and we take the max pooling operation on graphs for graph data. The output features are subjected to max pooling in order to obtain global features, and then, the classification results are obtained through the fully connected layer (512, 256, and 40) and the Softmax layer.

## 4. Experiments and Results

We conducted experiments to prove the usefulness of our classification model on ModelNet40 [10]. Table 1 shows the parameter settings for model training. We used the PyTorch 1.10.1 (Soumith Chintala, America) deep learning framework and Python 3.6 (Guido van Rossum, Holland). Hardware configuration: CPU: Intel Xeon Platinum 8259CL, GPU: Tesla T4 with 16 G of video memory.

Many research institutions have opened datasets with 3D models, and we will apply the model proposed in this paper on ModelNet40 for the classification. ModelNet40 has 40 different 3D shape models with 12,311 CAD models, including their mesh information, of which 9843 models are used as the training sets and 2468 models are used as the test sets. Before conducting the experiments, each model needs to be simplified to 1024 faces. If the number of faces is not enough, the existing faces are randomly selected to be filled, which normalizes the model to the unit ball centered at the origin.

Table 2 shows the experimental results of the model in this paper on the ModelNet40 dataset. The overall accuracy is the accuracy of the classification of all 3D models in the dataset, and the mean class accuracy is the average of the accuracy of each category. The overall accuracy of the proposed method compared with existing methods in this paper is 6.3% higher than 3D ShapeNets, 8% higher than VoxNet, and 0.3% higher than PointNet++. We proposed to make full use of the mesh data information by extracting the center points and corner vectors of the faces, which can enhance the ability to describe the local information by considering the spatial position of the faces in the 3D model, the structure of the faces themselves, and the adjacency relationship between the faces. The experimental results show that the model proposed in this paper effectively captures local information from triangular meshes. The classification accuracy of the model in this paper can achieve better results on the representation based on a triangular mesh.

**Table 2 sensors-22-07040-t002:** Comparison of classification results on ModelNet40.

Model	Representation	Overall Accuracy (%)	Mean Class Accuracy (%)
MVCNN [7]	view	90.1	79.5
VoxNet [21]	volume	83.0	85.9
PointNet [24]	point	89.2	86.2
PointNet++ [25]	point	90.7	-
Kd-Net [22]	point	90.6	88.5
SO-Net [36]	point	90.8	-
Momenet [37]	point	89.3	86.1
LKPO-GNN [38]	point	90.9	88.2
ReebGCN [39]	point	89.9	87.1
Ours	mesh	91.0	89.1

Furthermore, we also experimented with the model of this paper on the ShapeNet dataset [40], which has 16 classes with 16,881 shapes. For the ShapeNet dataset, the overall classification accuracy of this paper’s model is 90.8%, which has the potential to achieve good classification results.

An analysis of the spatial complexity and time complexity of the model is shown in Table 3. We used params (number of parameters) to represent the spatial complexity and FLOPs (floating operations conducted for each input sample) to express the time complexity. The classification method proposed in this paper uses only two layers of convolutional neural networks, thus reducing the number of parameters of the model. The point cloud-based classification task algorithm is the most efficient among other data types. The proposed method can achieve comparable results with the point cloud-based method in terms of spatial complexity and time complexity. Therefore, the method proposed in this paper is concise and effective.

Table 4 compares the classification effects of the model using GCNs with a different number of layers. The overall classification accuracy of the model was 90.2% when using one-layer GCN; the best performance was achieved when using two-layer GCN with an overall classification accuracy of 91%; and the classification ability of the model started to decrease when using three-layer GCN, with an overall classification accuracy of 86.8%. The performance of the model decreased again when using the four-layer GCN, with an overall classification accuracy of 83.6%. The essence of GCN is aggregating neighbor information. Every time the node’s features are updated, they aggregate the information of higher order neighbor nodes for any node in the graph. As the number of GCN layers increases, once a certain threshold is reached, the nodes covered by each node converge to the full graph node. This leads to a significant reduction in the diversity of the local network structure of each node, which is very detrimental to the nodes’ feature learning. Therefore, we use the two-layer GCN classification model in this paper.

To investigate the effect of the number of faces in the 3D model on the classification accuracy, we re-simplified the dataset to obtain four kinds of 3D model datasets, with the number of faces being 512, 1024, 2048, and 4096, and input them into the network for the classification experiments. We list the classification results as shown in Table 5. The experimental results show that an increase in the number of faces of the 3D model results in a little improvement in the accuracy of the classification. However, a large number of faces consumes a large amount of computer memory and computing time. To guarantee the network performance and computation speed, we choose to simplify the model to 1024 faces for the experiment.

To analyze the effectiveness of each module in our classification model, we designed ablation experiments to compare the accuracy of ModelNet40 classification under different combinations of modules. Center Module refers to the center point feature of the face and the subsequent fully connected layer. Corner Module refers to the corner vector feature of the face and the rotated convolution layer behind it. GCN refers to the graph convolutional neural network, and after removing the GCN, we pass the initial features directly through the fully connected layer. Table 6 shows the results of the ablation experiments. The experimental results show that only using grid features for classification is not effective, but the addition of the GCN module can help improve the transmission of information in the network and can capture local features better. Finally, we can obtain shape descriptors that have rich features.

## 5. Conclusions

In this paper, we proposed a 3D shape classification network that combines the features of mesh data and is capable of learning the mesh data directly. By examining the mesh data’s features and by generating an adjacency matrix based on nearby faces, the graph convolution can be applied to the mesh data. Our model is better able to capture 3D model features because of the benefits of using graph convolutional neural networks to analyze non-Euclidean data. The experimental results prove that the classification model in this paper is lighter in terms of the number of parameters and has a better classification effect.

## Figures and Tables

**Figure 1 sensors-22-07040-f001:**
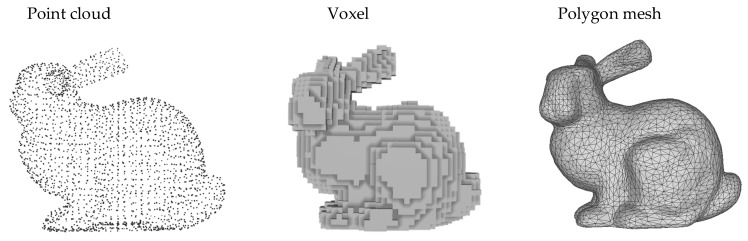
Point cloud, voxel, and polygon mesh representation of 3D models.

**Figure 2 sensors-22-07040-f002:**
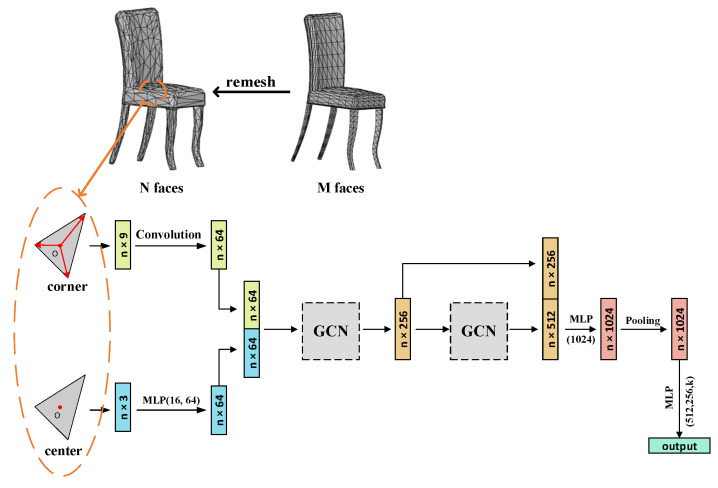
Algorithm flow chart. (After feature extraction of the faces, they are input to the classification network for aggregated vertex information and then pooled to obtain global features. Finally, the classification results are output.)

**Figure 3 sensors-22-07040-f003:**
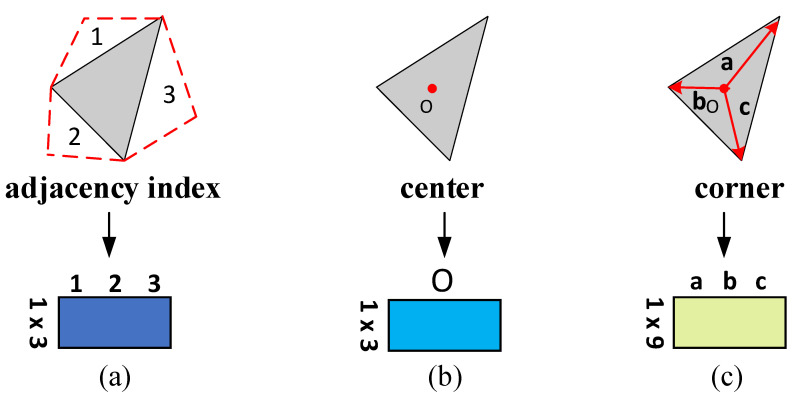
Initial feature extraction: (**a**) indexing of adjacent faces; (**b**) coordinates of the center point of the face; and (**c**) the vector from the center point to each of the three vertices. **a**, **b**, and **c** denote the vectors from the center point of the triangle to the vertices.

**Figure 4 sensors-22-07040-f004:**
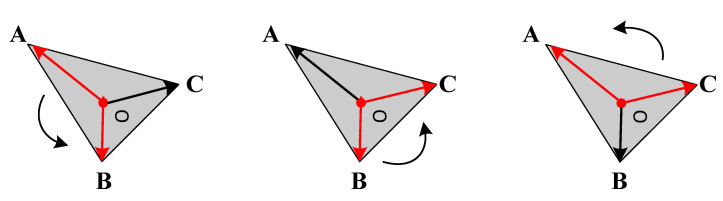
Rotational convolution schematic. **A**, **B**, **C** represent the three vertices of the triangle, O is the center point of the triangle, and the curved arrow outside the triangle represents two adjacent angle vectors for convolution operation.

**Figure 5 sensors-22-07040-f005:**
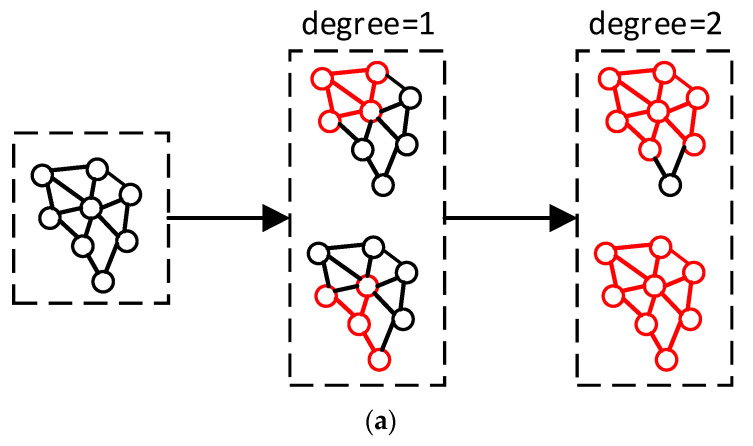
(**a**) Schematic diagram of two-layer graph convolution calculation. (**b**) Feature splicing schematic. Purple represents the information features of the aggregated first-degree neighboring vertices and blue represents the information features of the aggregated second-degree neighboring vertices.

**Table 1 sensors-22-07040-t001:** Training parameter setting.

Optimizer	Lr	Batch	Weight Decay	Epoch
ADAM	0.001	128	0.0005	250

**Table 3 sensors-22-07040-t003:** Space complexity and time complexity for classification.

Model	Params (M)	FLOPs (G)
PointNet [24]	3.48	0.44
PointNet++ [25]	1.48	-
Kd-NET [22]	7.44	-
MVCNN [7]	60.00	62.06
Ours	1.47	1.61

**Table 4 sensors-22-07040-t004:** Comparisons of overall accuracy when using a different number of GCN layers. We use the ModelNet40 dataset.

Model	GCN Layers			
	1	2	3	4
Ours	90.2%	91.0%	86.8%	83.6%

**Table 5 sensors-22-07040-t005:** Classification results with a different number of faces on ModelNet40.

Number of Faces	512	1024	2048	4096
Accuracy	90.3%	91.0%	90.8%	91.0%

**Table 6 sensors-22-07040-t006:** Classification results of ablation experiments on ModelNet40.

Center Module	√	√	√
Corner Module	√	√	
GCN	√		√
Accuracy	91.0%	89.6%	87.8%

## Data Availability

Not applicable.

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
