# Peer review of "Deep Neural Network for 3D Shape Classification Based on Mesh Feature"

_sensors, 2022, doi:10.3390/s22187040_

Round 1
Reviewer 1 Report
The paper deals with an interesting topic 3D similarity focused on deep learning. In the present form the paper needs improvements. Starting from the introduction authors should provide a wider picture on 3D morphometric analysis in order to better contestualise and justify the proposed methodology. Some more references should be added as for instance the following ones:
Ulrich, Luca, et al. "Can ADAS Distract Driver’s Attention? An RGB-D Camera and Deep Learning-Based Analysis." Applied Sciences 11.24 (2021): 11587.
Gao, Zan, Yinming Li, et al., deep learning for view-based 3D model retrieval." ACM Transactions on Multimedia Computing, Communications, and Applications (TOMM) 16.1 (2020): 1-21.
Regarding the methodological section authors should provide more details regarding the theoretical framework. For what concerns the experimental validation some more issue on the usages scenarios and relatives performs should be added
Reviewer 2 Report
This paper presents Deep Neural Network for 3D Shape Classification Based on Mesh Feature.
Comments to improve the manuscript are as follows; 1. Authors mention the term "3D shape" in the title, abstract, and the introduction. But, the experiments have been conducted only using the face 3D data. So, mentioning the term "3D shape" is not fair, since all the 3D shapes are not like faces. 2. The performance results of the proposal are almost the same as results as PointNet++ and LKPO-GNN[33]. So, what are the significant points in the proposal? How about the computational time of the proposal? Reviewers recommend adding a comparison regarding computational time of the proposed method and conventional method. 3. Reviewer believes that meshes generation may increase the learning performance in face detection. But, when you apply this method to the detection of 3d Objects having sharp edges, performance would be weak since you may dramatically lose the shape information. 4. Authors have not well searched literature regarding the 3D object detection. According to my observation the below latest papers could be found. Head Posture Estimation by Deep Learning Using 3-D Point Cloud Data From a Depth Sensor (same authors look have published more similar works), A Survey on 3D Point Cloud Compression Using Machine Learning Approaches.Author Response
Please see the attachment.

Round 2
Reviewer 2 Report
N/A
